# A novel oncolytic vaccinia virus with multiple gene modifications involved in viral replication and maturation increases safety for intravenous administration while maintaining proliferative potential in cancer cells

**Go Okita** [ID]*, **Kiyotaka Suenaga** [ID]¤, **Masashi Sakaguchi, Toshio Murakami**

Research Department, KM Biologics Co., Ltd., Kikuchi, Kumamoto, Japan

¤ Current address: Development Department, Meiji Animal Health Co., Ltd., Kumamoto, Japan
* okita-go@kmbiologics.com

## Abstract

To generate a novel oncolytic vaccinia virus with improved safety and productivity, the genome of smallpox vaccine strain LC16m8 was modified by a bacterial artificial chromosome system. By using LC16m8, a replicating virus homologous to the target virus, as a helper virus for the bacterial artificial chromosome system, we successfully recovered genome-edited infectious viruses. Oncolytic viruses with limited growth in normal cells were obtained by deleting the genes for vaccinia virus growth factor (VGF), extracellular signal-regulated kinase-activating protein (O1L), and ribonucleotide reductase (RNR) present in the viral genome. Furthermore, the amino acid residues of seven proteins involved in extracellular enveloped virus virion formation were replaced to the IHD-J strain sequence, which is known to highly express extracellular enveloped virus. In cultured cancer cells (HeLa), these modified viruses showed cytotoxicity and increased productivity, but it was confirmed that the cytotoxicity was suppressed in normal cells (normal human dermal fibroblasts). For *in vivo* safety evaluation, a modified virus (MD-RVV-ΔRR-EEV6) in which the VGF, O1L, and RNR genes of LC16m8 were deleted and the genes of six extracellular enveloped virus-associated proteins were replaced with sequences derived from IHD-J strain, and another modified virus (MD-RVV) lacking only the VGF and O1L were administered intravenously to severe combined immunodeficiency mice. In the MD-RVV administration, animals in all dose groups died by 40 days after virus administration. On the other hand, after MD-RVV-ΔRR-EEV6 administration, 3 out of 5 animals in the high and medium dose groups and all animals in the low dose group were still alive by day 71, the end of the observation period. These results demonstrate that genome editing of oncolytic vaccinia virus can delete genes involved in viral replication to improve safety in normal cells, while replacing genes involved in maturation improves proliferative potential in cancer cells.

**Data availability statement:** All relevant data are within the manuscript (DOCX) and figures (TIFF) files.

**Funding:** This work was fully supported by KM Biologics Co., Ltd. All authors were employees of KM Biologics Co., Ltd. at the time of this study. The founder had no role in the study design, data collection and analysis, decision to publish, or preparation of the manuscript.

**Competing interests:** GO and TM are employees of KM Biologics Co., Ltd. KS and MS are employees of Meiji Animal Health Co., Ltd. In connection with this research, KS, GO, and MS are the inventors of a registered patent (JP7034818B2), TM and GO are the inventors of an applied patent (WO/2021/029385). These does not alter our adherence to PLOS ONE policies on sharing data and materials.

## Introduction

Oncolytic virotherapy with various viruses represents a promising new therapeutic modality for cancer. Clinical trial data on the intravenous administration of oncolytic viruses demonstrated that they may be delivered safely and systemically with reduced toxicity; however, efficacy varied among individuals. This is primarily because at doses and regimens that are known to be safe, the virus is rapidly cleared from the circulation before it reaches its target. This phenomenon is mainly caused by neutralizing antibodies, complement activation, antiviral cytokines, and endogenous tissue macrophages, and is accompanied by non-specific uptake by other tissues, such as the lung, liver, and spleen [1].

Among oncolytic viruses, vaccinia virus has been the focus of preclinical and clinical research due to its many favorable properties [2]. The use of vaccinia virus in oncolytic virotherapy is expected to be effective not only for the treatment of primary tumors by intratumoral administration, but also for the treatment of metastatic cancers by systemic intravenous injection [3]. However, in a previous clinical trial, viral shedding-related adverse events were observed when oncolytic vaccinia virus was intravenously administered, concurrent with findings suggestive of efficacy [4,5]. Further improvements in the safety of oncolytic viruses for systemic administration have been sought.

The vaccinia virus complement control protein (VCP) secreted by vaccinia virus binds and inactivates complements C4b and C3b, thereby inhibiting the classical and alternative pathways of complement activation [6,7]. The vaccinia virus extracellular enveloped virus (EEV) incorporates host proteins into its membrane that exhibit decay-promoting or factor I cofactor activity and may prevent complement activation. The EEV protein A56R anchors secreted VCP to protect it from complement attack [8]. In addition, in studies with the IHD-J strain of vaccinia virus, the combined effects of the presence of a wrapping membrane and the process of internalization via an endocytic mechanism rendered EEV resistant to neutralizing antibodies [9]. The amounts of EEV that vaccinia virus strains release from infected cells markedly vary. The IHD-J strain has been shown to produce up to 100-fold more EEV than the related WR strain in RK13 cells [10].

Vaccinia virus utilizes the epidermal growth factor (EGF) receptor (EGFR) signaling pathway to promote its spread through the rapid and direct motility of infected cells [11]. The C11R protein, a vaccinia virus growth factor (VGF) that is similar to EGF and is secreted at an early stage of vaccinia virus infection, binds to EGF receptors on infected cells and surrounding cells to activate a signal through the MAP kinase cascade (the Ras/Raf/MEK/ERK metabolic pathway). In addition, the O1L protein of vaccinia virus constitutively activates extracellular signal-regulated kinase (ERK) in infected cells and promotes the pathogenicity of the virus [12]. Mitogen-activated protein kinase (MAPK)-dependent recombinant vaccinia virus (MD-RVV) with deletions in both VGF and O1L inhibits viral growth because ERK cannot be activated in normal cells. However, in cancer cells with abnormally activated MAPK pathways due to oncogenic mutations in the *EGFR*, *Ras*, and *Raf* genes, the deleted ERK activation function is complemented, and the virus proliferates and becomes an oncolytic virus that destroys tumor cells [13].

The vaccinia virus genome has two genes encoding ribonucleotide reductase (RNR), which is a rate-limiting enzyme in DNA synthesis and includes I4L (large subunit; RRM1) and F4L (small subunit; RRM2). Poxvirus RRM2 is required for efficient replication in cultured cells and virus toxicity in mice and forms functional complexes with host RRM1 to provide sufficient dNTPs for viral replication [14]. The safety of F4L-deleted oncolytic viruses has also been demonstrated using a bladder cancer model [15]. Based on these findings, we expect the further deletion of F4L from MD-RVV, in which C11R and O1L functions were blocked, to significantly suppress viral growth in normal cells and further improve safety.

The inhibited proliferation of oncolytic virus in normal cells is often accompanied by decreases in viral replication in cancer cells [16]. Attempts to compensate for the decline in efficacy and productivity caused by genetic modifications to the viral genome may increase productivity while maintaining the safety of oncolytic viruses. Replication-suppressed virus production may be restored by increasing infectivity. To promote the proliferative capacity of viruses that lack the functions of ERK and RNR involved in cancer cell proliferation, we focused on virion formation-associated molecules that are expressed late in the viral life cycle and involved in infectivity. Four different forms of vaccinia virus have been identified: intra-cellular mature virus (IMV), intracellular enveloped virus (IEV), cell-associated enveloped virus (CEV), and EEV. In the present study, the MD-RVV gene sequences encoding seven proteins (A33R, A34R, A36R, A56R, B5R, F12L, and F13L) involved in EEV formation were replaced with those of the IHD-J strain.

A method involving a bacterial artificial chromosome (BAC) system is used to modify the vaccinia virus genome. When a homologous poxvirus is used as a helper virus to reactivate another poxvirus, there is a risk of DNA recombination between the two. Previous studies have used a helper non-replicating fowlpox virus to recover the modified vaccinia virus-BAC plasmid in mammalian cells as an infectious virus [17]. In another attempt, cells infected with a non-replicating psoralen-UV-inactivated helper virus were used to generate recombinant vaccinia virus. It was shown that any poxvirus strain, including vaccinia virus itself can be used as a helper virus for the production of recombinant viruses [18]. In this study, we recovered infectious virus without using a heterologous virus by using a recombinant virus derived from the replicating vaccine strain LC16m8 genome as a helper virus.

## Materials and methods

### Cells

The immortalized cell line HeLa derived from human cervical cancer cells and the cell line RK13 exhibiting an epithelial morphology that was isolated from the rabbit kidney were pur-chased from the American Type Culture Collection. Primary normal human dermal fibro-blasts (NHDF) isolated from the dermis of adult skin were purchased from Lonza. These cells were generally handled according to the supplier's recommendations.

### Animals

Five-week-old female severe combined immunodeficiency (SCID) mice (C.B-17/IcrHsd-Prkdcscid) were purchased from Japan SLC. The protocol was approved by the KM Biologics Institutional Animal Care and Use Committee (approval number: C18-232) and was per-formed between December 4, 2018 and February 21, 2019. For animal welfare considerations, individual identification of the 50 mice was performed by dye application, and the number of mice housed per cage was 5. Animal health and behavior were monitored approximately every two days.

In evaluating the number of survivors, mice that showed symptoms of viremia were con-sidered dead if they did not move for a certain period of time and respiratory arrest and car-diac arrest were observed. As a humane endpoint, animals were euthanized by carbon dioxide inhalation if they had difficulty eating and drinking water, had a 30% weight loss compared to when they were accepted, and were judged to be at high risk of death due to debilitation. Once animals reached endpoint criteria, they were immediately euthanized. Some animals died before criteria for euthanasia were confirmed. Euthanasia during the observation period and after the end of the test was performed in consideration of the flow rate of carbon dioxide gas and the density of animals.

## Modifying the genome of vaccine strain vaccinia virus

The genome of LC16m8 vaccinia virus, a smallpox vaccine strain that has been administered to humans [19], was modified to produce recombinant vaccinia viruses (Fig 1).

## Construction of modified BACmid

The vaccinia virus genome (GenBank: AY678275.1) was modified using the BAC system [20].

In the first step, an insertion plasmid (pUC-VVTK-BAC-EGFP) was introduced by electroporation into rabbit primary kidney cells (PRK; prepared in-house) infected with vaccinia virus strain LC16m8 (prepared in-house). The expression of enhanced green fluorescent protein (EGFP) was observed using a fluorescence microscope, and wells with a high dilution ratio yielding fluorescence-positive plaques or cytopathic effects were selected. Cells in selected wells were collected, and the supernatant after sonication and centrifugation was obtained as virus (LC16m8-BACgfp). After infecting PRK with the BAC virus, pCAGGS-Cre plasmid (prepared in-house) was transfected using a transfection reagent. LC16m8-BACmid was then extracted as a circularized viral genome, electroporated into *Escherichia coli* (*E. coli*) strain GS1783 (NU2950) licensed from Northwestern University, *E. coli* harboring LC16m8-BACmid was selected, and target clones were obtained [21]. Each modified BACmid was prepared using LC16m8-BACmid as a template. The LC16m8 strain is an attenuated virus that may be used as a safe vaccine strain. However, from the perspective of the effectiveness of oncolytic viruses, it is preferable to use viruses that have high propagation properties within tumor cells. LC16m8 is known to produce an incomplete B5R protein due to a frameshift caused by a single nucleotide deletion in the wild-type B5R gene sequence, resulting in reduced viral propagation. In this study, a B5R modification cassette was prepared to modify the B5R gene sequence of LC16m8-BACmid to the complete B5R gene sequence of the parent strain LC16mO [19].

The C11R deletion cassette was constructed by deleting 255 bp from the start codon of the C11R gene sequence to the restriction enzyme AccI site to create a virus lacking VGF function. An O1L deletion cassette was prepared by deleting 1049 bp from the start codon of the O1L gene sequence to the restriction enzyme XbaI site, and inserting the kanamycin resistance gene sequence 50 bp immediately after the XbaI site to create a virus lacking O1L function. The O1L deletion cassette was obtained by treating this plasmid with the restriction enzymes ScaI and EcoRI. The F4L deletion cassette was prepared by deleting 765 bp from the start codon of the F4L gene sequence to the EcoRI site and, inserting the kanamycin resistance gene sequence 50 bp immediately after the EcoRI site to create a virus lacking RNR function. pUC57-ΔF4L-rKanI was treated with the restriction enzymes BamHI and HindIII and used as an F4L deletion cassette to modify BACmid.

Cassettes to modify the A33R, A34R, A36R, A56R, B5R, F12L, and F13L genes were constructed with the aim of generating viruses in which EEV-related genes from LC16m8 were replaced with the corresponding DNA sequences of the IHD-J (except for F12L) and IHD-W (F12L) strains (GenBank: AB191187.1, AB191188.1, AB191189.1, AB191190.1, KJ125439.1, and AB191191.1) [19,22]. After each EEV-related gene was replaced with the IHD-J or IHD-W sequence, artificially synthesized plasmids pUC57-A33R-rKanI, pUC57-A34R-rKanI, pUC57-A36R-rKanI, pUC57-A33-34-36R-rKanI, pUC57-A56R-rKanI, pUC57-B5R-rKanI, and pUC57-F12-13L-rKanI were constructed into which the kanamycin resistance gene sequence was inserted. They were then treated with restriction enzymes and used as EEV-related genes modification cassettes.

*E. coli* harboring LC16m8-BACmid was cultured, and the B5R-modified cassette was introduced by electroporation. Thereafter, the C11R deletion cassette, the O1L deletion cassette,

**Fig 1. Genome structure diagram of recombinant viruses.** Various oncolytic viruses were obtained by genetically modifying the viral genome of the vaccine strain LC16m8 as a starting material. The gene nomenclature used in this paper is that used for the vaccinia virus Copenhagen strain. Genes associated with viral attenuation and EEV productivity are indicated by dashed lines and black arrows, respectively. Black arrows are genes from the IHD-J (except for F12L) and IHD-W (F12L) strains. Env means the genes of extracellular enveloped virus-associated proteins.

and the F4L deletion cassette were introduced in this order. EEV-related gene modification cassettes were also introduced in the same manner. After selectively culturing the electroporated *E. coli* with appropriate agents, the modified BACmids were extracted. The band size was confirmed by PCR using primer pairs specific to the modified BACmids and nucleotide sequence analyses using sequencing primers were performed to confirm that the resulting clones were the desired modified BACmids (Table 1). Furthermore, a cassette (BACgfp removal sequence cassette) for removing the BACgfp sequence from the target virus was prepared and introduced into the modified BACmids described above.

## Production of an infectious modified virus

Since modified BACmids do not form virus particles by themselves, infectious viruses derived from modified BACmids can be obtained by simultaneously infecting cells with a helper virus and using enzymes necessary for virus production derived from the helper virus. In this study, using LC16m8-BACgfp as a helper virus, LC16m8-B5RmO-SBTKdup-BACmid-derived virus was recovered from RK13 cells. Cells were collected under a fluorescence microscope from wells with a high proportion of EGFP fluorescence-negative plaques, and after freeze-thawing and sonication, the centrifuged supernatant was obtained as a virus solution. These operations were repeated until all virus plaques in RK13 cells became fluorescently negative twice in a row, and the resulting virus was named LC16m8-B5RmO. Viruses derived from

**Table 1. Primers used to confirm modified BACmids and MD-RVV-ΔRR-EEV6.**

| Region | Confirmation method[a] | | Primer type | Primer sequence (5'-3') |
|---|---|---|---|---|
| | PCR | Sequencing[b] | | |
| C11R | * | ** | Forward | agaaccagctgctccatgatt |
| | * | | Reverse | gatacggaaccacccactgt |
| O1L | * | ** | Forward | tgtcaacggaccccaacatc |
| | * | | Reverse | acatggacgcattgggtgat |
| F4L | * | ** | Forward | caaggtctagacaaaccctcgt |
| | * | | Reverse | gcttcccacaacaatctcgc |
| A33R | * | * | Forward | aggttgcgtctagcactacg |
| | * | | Reverse | tgtatccatccattggcgca |
| A34R | * | * | Forward | aagcaaattgcactgcggaa |
| | * | | Reverse | cggtcaccgtgataagaggt |
| A36R | * | * | Forward | agacggcaatggatggatca |
| | * | | Reverse | tacaacgtgacggcagcaat |
| A56R | * | * | Forward | aatcccgctctatggtcagc |
| | * | | Reverse | agaaagtttccggcggctat |
| | | * | Forward | tctccatacgatgatctagt |
| B5R | * | * | Forward | tgcgtactacctgctgttgt |
| | * | | Reverse | aaatgctctaacggcatcgt |
| F12-13L | * | * | Forward | tggatacgaagatgctatccatca |
| | * | | Reverse | ccaattcctatgtctagatt |
| | | * | Forward | tgtactagtcataatatctt |
| | | * | Forward | aatctagacataggaattgg |
| | | * | Forward | ctctattaatggctgcttct |

[a]The confirmation method performed with that primer is indicated by the * mark.

[b]Primers used only for the sequence of MD-RVV-ΔRR-EEV6 are indicated by the ** mark.

other modified BACmids were prepared in the same method as above, and named MD-RVV, MD-RVV-ΔRR, MD-RVV-A34R, MD-RVV-ΔRR-A34R, MD-RVV-EEV6, MD-RVV-ΔRR-EEV6, MD-RVV-EEV7, and MD-RVV-ΔRR-EEV7 (Fig 1).

MD-RVV-ΔRR-EEV6, the infectious virus used for evaluation in this study, was cultured in a bioreactor (iCELLis nano, Pall) for 2 days using HeLa as host cells, and then DNA was extracted. Using the primers in Table 1, DNA in the modified region was amplified and the partial genome sequence was analyzed.

## Viral cytotoxicity

The *in vitro* cytotoxicity of the modified virus in cancer and normal cells was investigated. Cancer cells (human cervical carcinoma-derived cell line: HeLa) and normal cells (NHDF) were seeded on 96-well plates ($0.5 \times 10^4$ cells/well) and subjected to an adhesion culture in a serum-containing medium. Each cell was then serum-starved in serum-free Eagle's minimum essential medium and inoculated with low ($8.0 \times 10^4$ PFU/well) and high ($4.0 \times 10^5$ PFU/well) levels of the modified viruses. After culturing for 72 hours, the cell viability of each virus-inoculated group was quantified using Cell Counting Kit-8 (CCK-8; Dojindo Laboratories) with the absorbance of the control group without virus inoculation set to 100%. CCK-8 contains a tetrazolium salt (WST-8) and an electron carrier (1-Methoxy PMS). CCK-8 solution was placed in each well of the cultured plates, and the absorbance at 450 nm was measured after 2 hours of incubation. Under these conditions, it was confirmed that there was a linear correlation between the number of viable cells and the absorbance.

## Statistical analysis

To assess the reduced cytotoxicity of the engineered viruses (LC16m8-B5RmO, MD-RVV, and MD-RVV-ΔRR), the difference between the average values of triplicate measurement data of cell viability was evaluated by two-tailed paired Student's t-test using Microsoft Excel in Microsoft 365 MSO (version 2310, build 16.0.16924.20054, 64 bit).

Survival analysis was performed using the Kaplan-Meier method to estimate survival curves. The log-rank test was employed to compare survival distributions between groups. Statistical analyses were conducted using JMP software (version 17.2.0).

## Viral productivity in cancer cells

The productivity of the modified virus in cancer cells was investigated. HeLa cells were seeded on a 24-well plate, adherently cultured in a serum-containing medium, and then inoculated with the modified virus at MOI 1 PFU/cell. After culturing for 1 hour, the virus-containing medium was removed, and the serum-containing medium was newly added. After 16, 24, 32, and 48 hours, the medium was collected to obtain the virus in the culture supernatant. The recovered virus was serially diluted and inoculated into RK13 cells cultured in a 6-well plate. After culturing for 1 hour, the virus-containing medium was removed, methylcellulose medium was newly added, and after culturing at 37°C for 72 hours, plaques were counted to assess the infectivity titer.

In addition, HeLa cells were adherently cultured in a serum-containing medium and then inoculated with the modified virus in a chemically defined serum-free medium at MOI 0.01 PFU/cell. After culturing for 1 hour, the virus-containing medium was removed, and the serum-free medium was newly added. Culture supernatant viruses were collected after 48 hours. Furthermore, cells were collected by pipetting, frozen and thawed, then sonicated, and the supernatant after centrifugation was obtained as the intracellular virus. The infectivity titers of these specimens were assessed by the method described above.

### Intravenous administration to immunodeficient mice

The modified virus was intravenously administered once in 0.1 ml to immunodeficient mice (SCID mice, 5-week-old, female, 5 mice in each group) at doses of $2.5 \times 10^4$, $5.0 \times 10^5$, and $1.0 \times 10^7$ PFU/mouse. As a control group, the same volume of phosphate-buffered saline was administered. Survival rates, body weights, and viral symptom scores were observed. The virus symptom score is a maximum of 5 points as follows: 0: no symptoms; 1: 1 or 2 tail lesion(s); 2: 3 or 4 tail lesions; 3: more than 4 tail lesions, a rough coat, piloerection; 4: dyspnea, moribund; 5: death.

## Results

### Decrease in cytotoxicity in normal cells by the RNR gene deficiency

Cancer cells (HeLa) or normal cells (NHDF) were infected with LC16m8-B5RmO, in which the B5R gene of LC16m8 was reverted to the sequence of the parent strain (LC16mO), and its modified viruses under serum-free culture conditions, and the cytotoxicity of these viruses was compared. The modified viruses used were MD-RVV lacking the C11R and O1L genes of the LC16m8-B5RmO virus genome, MD-RVV-ΔRR lacking the F4L gene encoding the RNR small subunit of MD-RVV, and viruses that modified the EEV-related gene of MD-RVV-ΔRR (MD-RVV-ΔRR-A34R, MD-RVV-ΔRR-EEV6, and MD-RVV-ΔRR-EEV7).

In the process of preparing these modified viruses, we examined the growth of recombinant viruses that express EGFP integrated into the viral genome and emit fluorescence. HeLa cells were infected with LC16m8-BACgfp, LC16m8-B5RmO-BACgfp, MD-RVV-BACgfp, and MD-RVV-ΔRR-BACgfp at MOI 1 PFU/cell, virus-containing medium was removed, virus-free medium was added, and cells were then cultured for 24 and 48 hours. The EGFP fluorescence intensity of LC16m8-BACgfp was low, and the RNR-deficient virus (MD-RVV-ΔRR-BACgfp) showed fluorescence intensity equivalent to that of its parent strains, LC16m8-B5RmO-BACgfp and MD-RVV-BACgfp (Fig 2). No fluorescence was observed in mock HeLa cells that do not express EGFP. The partial genome sequence of MD-RVV-ΔRR-EEV6, the modified virus primarily used in the evaluation of this study, was confirmed by PCR and DNA

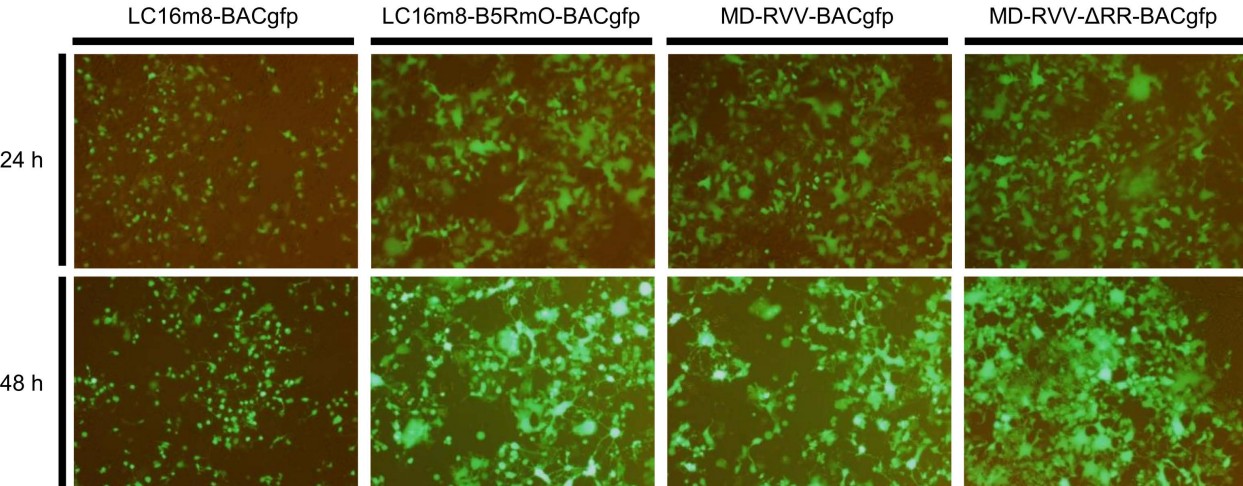

**Fig 2. Growth of F4L-deficient (ΔRR) virus in HeLa cells.** HeLa cells were inoculated with LC16m8-BACgfp, LC16m8-B5RmO-BACgfp, MD-RVV-BACgfp, and MD-RVV-ΔRR-BACgfp at MOI 1 PFU/cell. After 24 and 48 hours, the growth of each virus was observed under a fluorescence microscope.

sequencing using the primers listed in Table 1. The deleted region in the genome of this modified virus and the base sequences of the inserted gene matched the designed sequence.

All modified viruses used here exhibited similar cytotoxicity against HeLa cells, indicated by a reduction in the number of viable cells (Fig 3A). On the other hand, LC16m8-B5RmO and MD-RVV exhibited dose-dependent cytotoxicity against NHDF; however, inoculation with the F4L gene-deficient virus (MD-RVV-ΔRR) maintained a significantly (p < 0.005) higher viable cell rate compared to these two viruses (Fig 3B).

## Increasing productivity in cancer cells by modifying EEV-related genes

HeLa cells were infected with MD-RVV, MD-RVV-A34R, MD-RVV-EEV6, and MD-RVV-EEV7, virus-containing medium was removed, virus-free medium was added, and cells were then cultured for 16, 24, 32, and 48 hours. These culture supernatants were collected, and virus infectivity titers were assessed. The production of the virus in which the EEV-related gene was replaced with the sequence of the IHD-J strain increased over time. After 48 hours, MD-RVV-EEV7, which replaced all seven EEV-related genes, showed the most significant increase in production. The expression of A34R in the IHD-J strain is known to promote EEV production [10]. However, MD-RVV-EEV6, in which 6 types of genes other than A34R were replaced, showed an increase in virus production over that of MD-RVV-A34R, in which only A34R was replaced. Similar to these results, an increase in the production of the

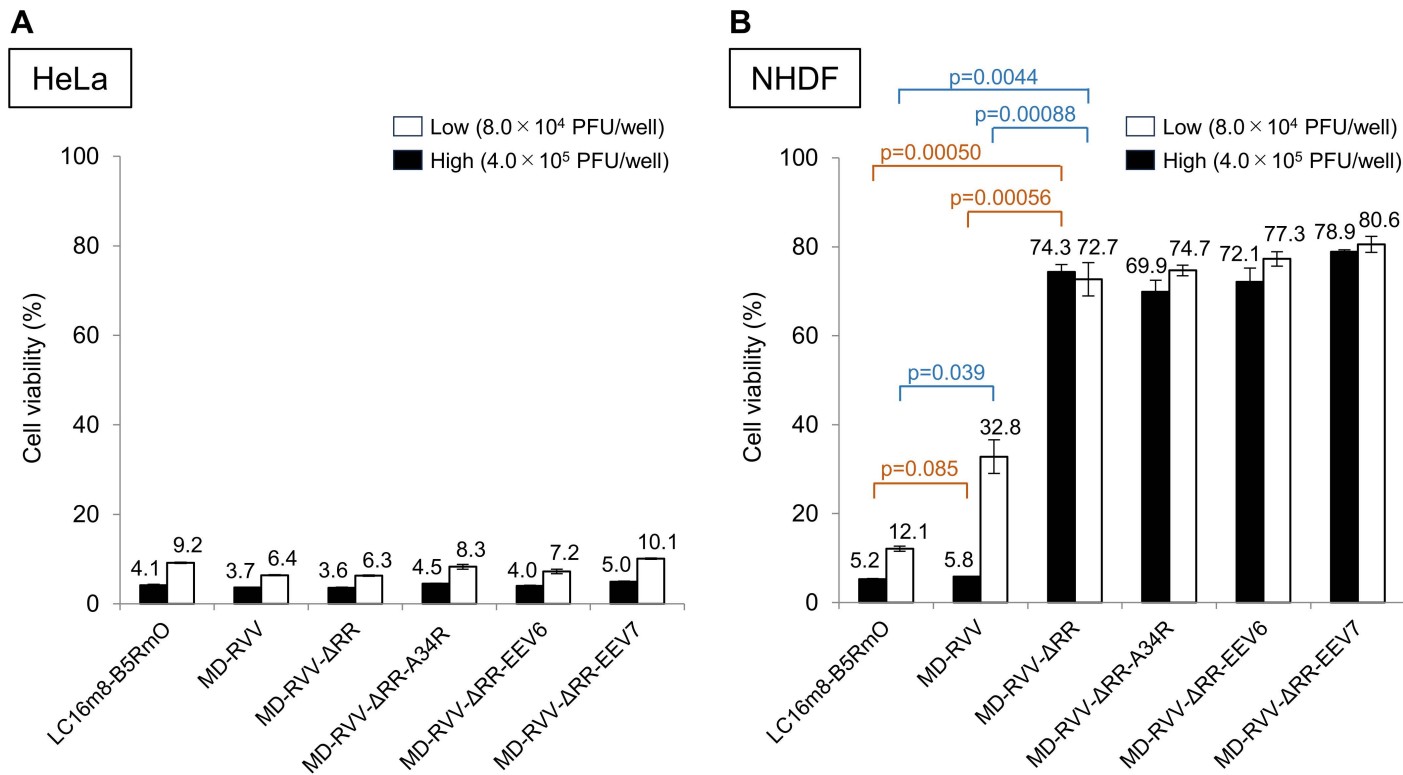

**Fig 3. Effects of F4L-deficient (ΔRR) modified viruses on the cytotoxicity of cancer cells and normal cells.** Cancer cells (**A**; HeLa) and normal cells (**B**; NHDF) were inoculated with $8.0 \times 10^4$ (white) and $4.0 \times 10^5$ (black) PFU/well of the indicated modified viruses. After culturing for 72 hours, the cell viability of each virus-inoculated group was quantified using a cell counting kit, with the absorbance of the control group without the virus inoculation being set to 100%. Data are shown the mean ± SEM (n = 3). Statistical significance for between LC16m8-B5RmO, MD-RVV, and MD-RVV-ΔRR in experiments using NHDF is indicated by two-tailed paired Student's t-test p-values (**B**).

MD-RVV-ΔRR-EEV6 and MD-RVV-ΔRR-EEV7 viruses was observed in viruses modified with the EEV-related genes of MD-RVV lacking the RNR gene F4L (Fig 4).

Regarding virus productivity in the above cancer cells, intracellular virus productivity was investigated together with the extracellular virus in the culture supernatant. After infecting HeLa cells with LC16m8-B5RmO, MD-RVV-ΔRR, MD-RVV-ΔRR-A34R, MD-RVV-ΔRR-EEV6, or MD-RVV-ΔRR-EEV7, the virus-containing medium was removed, and infected cells were cultured in virus-free medium for 48 hours. MD-RVV-ΔRR-EEV7 showed very high virus productivity in the culture supernatant, while that of MD-RVV-ΔRR-EEV6 was similar to those of LC16m8-B5RmO and MD-RVV-ΔRR-A34R (Fig 5A). On the other hand, the intracellular viral load of MD-RVV-ΔRR-EEV6 was as high as those of LC16m8-B5RmO and MD-RVV-ΔRR-EEV7 (Fig 5B).

## Improving the safety of the modified virus with the RNR deficiency in immunodeficient mice

To confirm the viral toxicity of the modified viruses generated in the present study, LC16m8-B5RmO, MD-RVV, and MD-RVV-ΔRR-EEV6 were intravenously injected once into SCID mice at $2.5 \times 10^4$ (low dose), $5.0 \times 10^5$ (medium dose), and $1.0 \times 10^7$ (high dose) PFU per mouse. The survival rate decreased in a virus dose-dependent manner, and animals in all dose groups died 19 and 40 days after virus administration in the LC16m8-B5RmO and MD-RVV

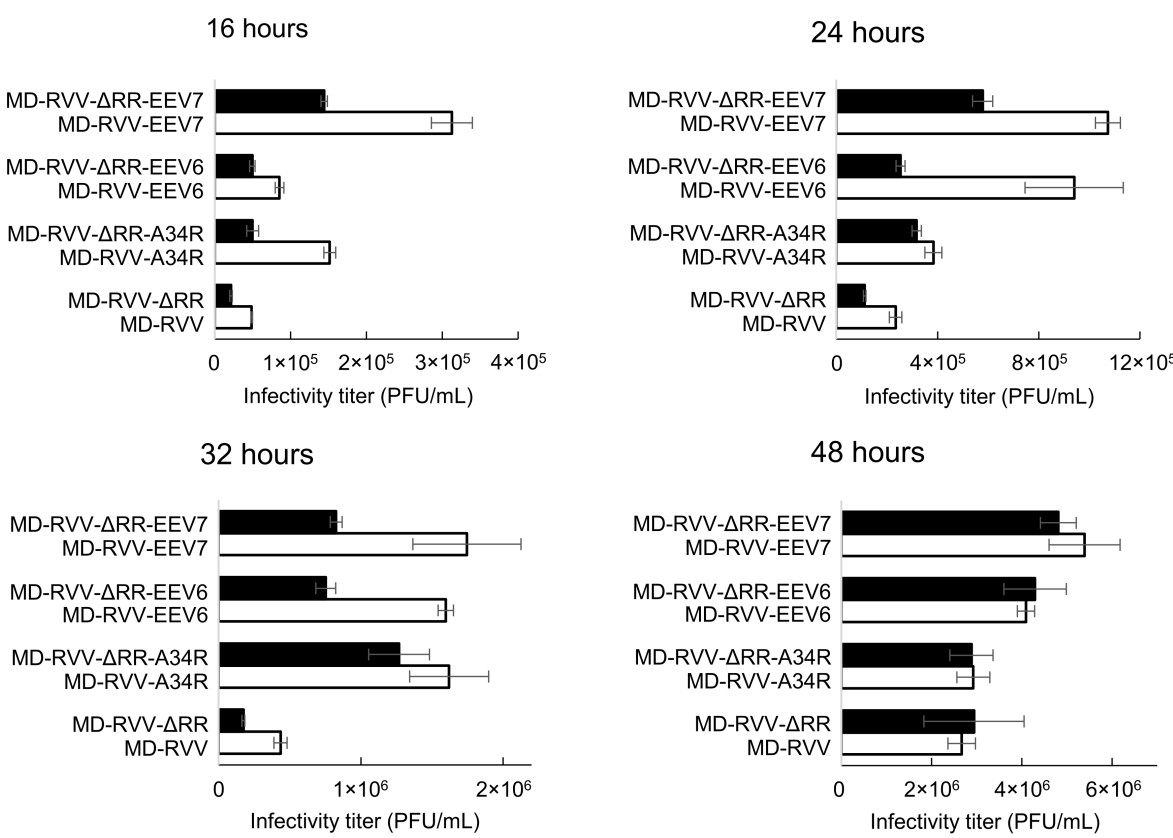

**Fig 4. Effects of the modification of EEV-related genes on viral productivity in HeLa cells.** HeLa cells were inoculated with the modified virus without (white) and with (black) the deletion of F4L (ΔRR) at MOI 1 PFU/cell. After 16, 24, 32, and 48 hours, the infectivity titer of the viruses was evaluated using RK13 cells. Data are shown the mean ± SEM (n = 2).

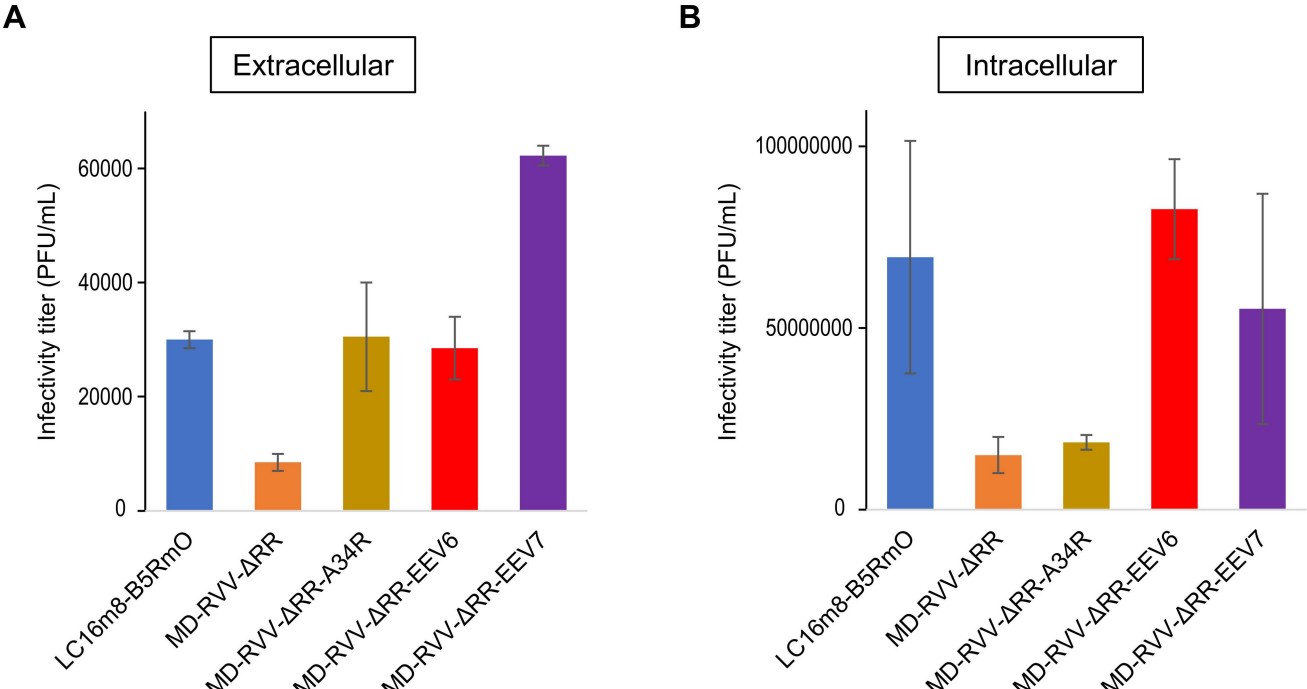

**Fig 5. Increased production of extracellular and intracellular viruses by modifying EEV-related genes.** HeLa cells were adherently cultured in a serum-containing medium and then inoculated with the modified virus in a chemically defined medium at MOI 0.01 PFU/cell. After 48 hours of culture, the extracellular virus in the culture supernatant **(A)** and intracellular **(B)** viruses were collected. The infectivity titers of the viruses were evaluated using RK13 cells. Data are shown the mean ± SEM (n = 2).

administration groups, respectively. On the other hand, after the administration of MD-RVV-ΔRR-EEV6, 3 out of 5 animals in the high and medium dose groups and all animals in the low dose group survived until day 71, the end of the observation period. The Kaplan-Meier survival curves for the two groups are shown in Fig 6. Log-rank tests between the two groups showed that MD-RVV-ΔRR-EEV6 had a significant difference in survival at all doses (p < 0.004).

Dose-dependent weight loss was observed in all mice administered to each virus. The degree of weight loss was greater in the order of LC16m8-B5RmO, MD-RVV, and MD-RVV-ΔRR-EEV6 (Fig 7). Regarding the virus symptom score observed as toxicity associated with virus administration, the change over time in the total score of each administration group was similar to that in body weight (Fig 8). At high administration doses of LC16m8-B5RmO, MD-RVV, and MD-RVV-ΔRR-EEV6, 1 to 4 pock(s) (score 1 or 2) were initially observed in all 5 animals in each group after 5, 7, and 13 days, respectively. At low administration doses, symptom scores of 1 or 2 were observed in all mice in the LC16m8-B5RmO and MD-RVV administration groups on days 7 and 13 after administration, respectively. However, even on day 70 after the administration of MD-RVV-ΔRR-EEV6, all mice survived with only 1 out of 5 having a symptom score of 2. Fig 9 shows photographs of the symptoms observed in representative individuals in each group at doses of $5.0 \times 10^5$ PFU/mouse on day 15 after administration.

## Discussion

Novel oncolytic vaccinia viruses derived from the vaccine strain LC16m8 virus genome were examined in the present study. The virus genome was modified using a BAC system. BAC

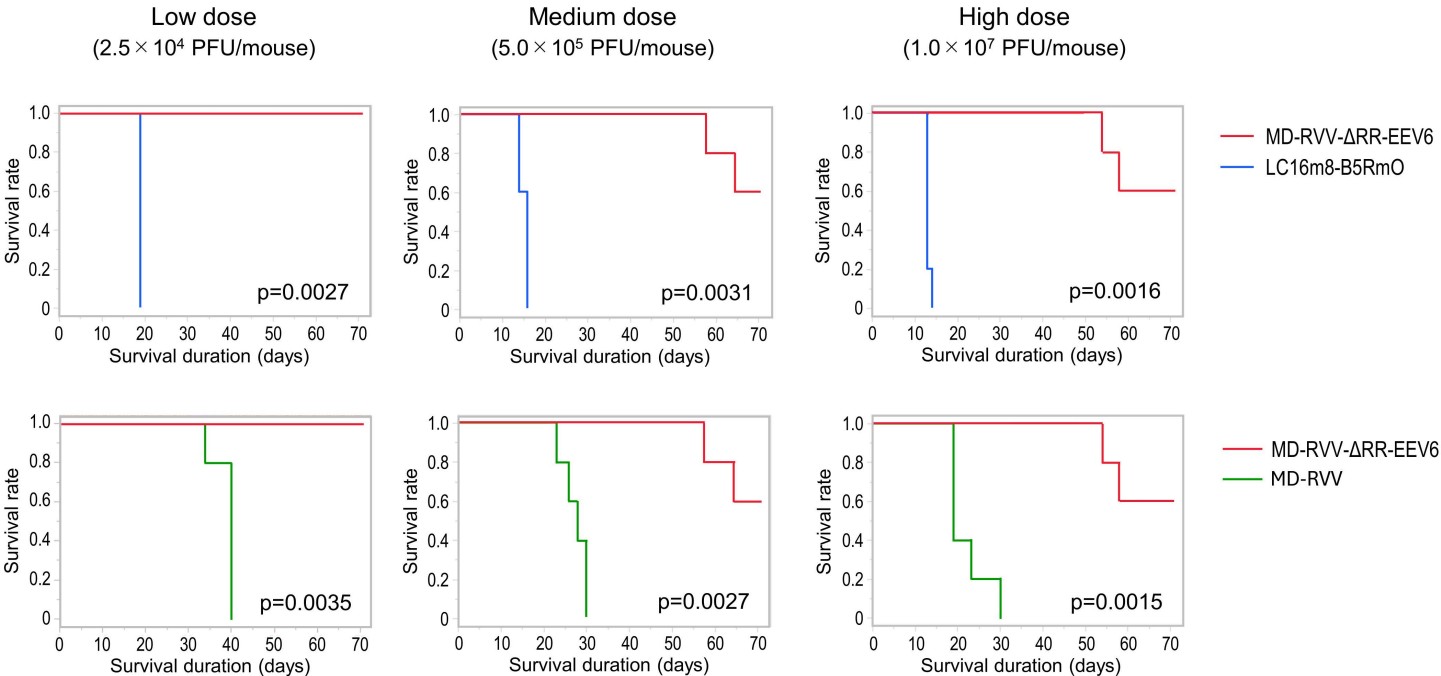

**Fig 6. Survival rates of immunodeficient mice intravenously administered modified viruses.** The modified viruses LC16m8-B5RmO (blue), MD-RVV (green), and MD-RVV-ΔRR-EEV6 (red) were intravenously administered once to five immunodeficient mice in each group at doses of $2.5 \times 10^4$ (Low), $5.0 \times 10^5$ (Medium), and $1.0 \times 10^7$ (High) PFU/mouse. The log-rank test was used to compare survival distributions between the indicated groups to determine p-values.

technology has made it possible to exploit the genetics of *E. coli* to precisely introduce multiple genetic mutations into the viral genome [17]. The rescue of infectious genome-edited vaccinia viruses has so far been consistently achieved by transfecting mammalian cells infected with a helper non-replicating fowlpox virus with a BAC plasmid with a vaccinia virus insert. The risks associated with using fowlpox as a helper virus include the possibility of contamination with viruses other than the vaccinia virus of interest and unintended recombination between the virus of interest and the helper virus. Using the vaccine strain LC16m8, a replicating virus homologous to the target virus, as a helper virus, we successfully recovered genome-edited infectious viruses. Furthermore, this study revealed that it is also possible to use a virus lacking F4L as a helper virus to suppress the proliferation of LC16m8. When producing modified viruses as pharmaceuticals in the future, it is important to use whole genome sequencing to confirm that there are no mutations in the gene sequence from virus seed to the end of virus culture.

The cancer cell-specific cytotoxicity of genetically modified vaccinia viruses was investigated using cancer cells (HeLa) and normal cells (NHDF). To avoid the effects of serum growth factors, each cell was cultured adherently in a serum-containing medium and then under serum starvation. The respective cells were inoculated with a low level ($8.0 \times 10^5$ PFU/ml) or high level ($4.0 \times 10^6$ PFU/ml) of each virus selected from LC16m8-B5RmO, MD-RVV, MD-RVV-ΔRR, MD-RVV-ΔRR-A34R, MD-RVV-ΔRR-EEV6, or MD-RVV-ΔRR-EEV7. All vaccinia viruses tested were also dose-dependently cytotoxic to HeLa. The infection of NHDF with MD-RVV and MD-RVV-ΔRR revealed that the deletion of the RNR gene significantly reduced cytotoxicity in NHDF. The suppression of cytotoxicity in NHDF by these RNR-deleted viruses was not affected by the modification of envelope-associated genes (Fig 3B). The up-regulated expression of RNR in cells is characteristic of many cancers, and

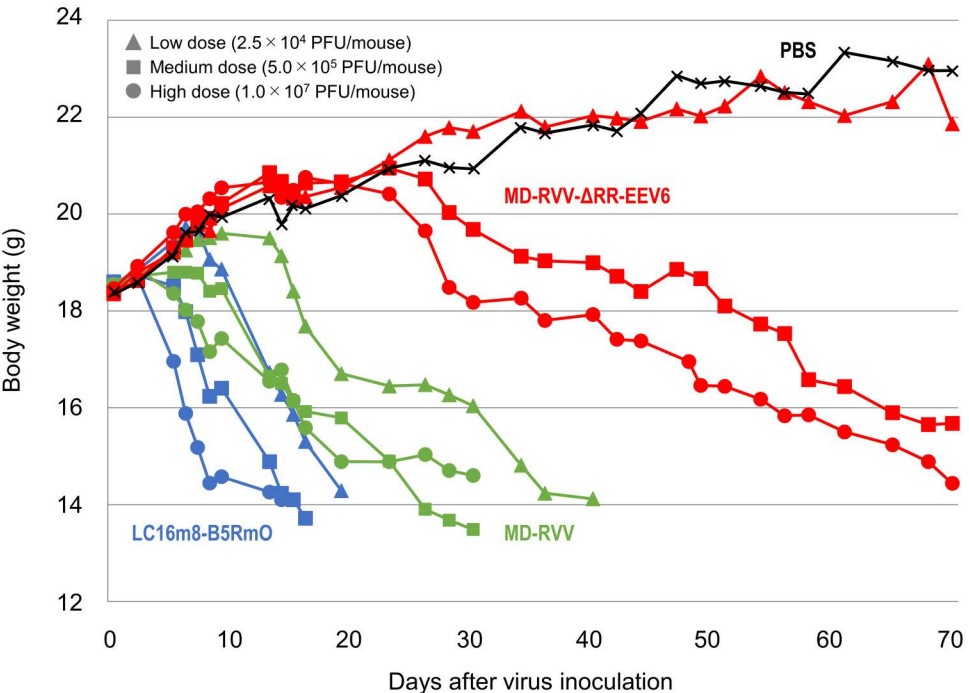

**Fig 7. Body weight changes in immunodeficient mice intravenously administered modified viruses.** The modified viruses LC16m8-B5RmO (blue), MD-RVV (green), and MD-RVV-ΔRR-EEV6 (red) were intravenously administered once to five immunodeficient mice in each group at doses of $2.5 \times 10^4$ (triangle marks), $5.0 \times 10^5$ (square marks), and $1.0 \times 10^7$ (round marks) PFU/mouse. As a control group (black; cross marks), the same volume of phosphate-buffered saline was administered. Body weights are expressed as the mean values of surviving mice.

an investigation of RNR gene expression in human cancers using the ONCOMINE database showed that RRM2 was ranked in the top 10% of the most overexpressed genes in 73 out of the 168 cancers analyzed [23]. In addition, RNR activity in mammals is dependent on the cell cycle, and the protein level of RRM1 remains constant throughout the cell cycle, while RRM2 is expressed in the G1/S phase during DNA replication. RNR activity is controlled during the cell cycle by RRM2 protein levels [24]. In contrast, the progression of the cell cycle from the G1 phase to the S phase is triggered by the activated ERK pathway [25]. Accordingly, vaccinia viruses with inhibited F4L functions may replicate using cell-derived RRM2 in cancer cells where cells are actively proliferating. However, the expression level of RRM2 in quiescent normal cells is low; therefore, F4L-deficient viral replication is restricted.

Although monoclonal antibodies against EGFR and inhibitors of the MAP kinase cascade have been developed as molecular targeted drugs, these anticancer drugs have side effects and drug resistance may develop [26,27]. The enhancement of the MAP kinase cascade and the expression of RRM2 associated with cell proliferation due to genetic mutations in the EGFR signaling pathway are favorable for the viral replication of MD-RVV-ΔRR in which ERK activation and RRM2 expression are suppressed. Therefore, MD-RVV-ΔRR is also expected to be effective against drug resistance to EGFR and MAP kinase cascade inhibitors.

The intravenous administration of an oncolytic virus has the advantages of convenience and speed, which are more suitable at the clinical trial stage [28]. Cells infected with vaccinia virus produce four viral forms that play distinct roles in the viral life cycle. IMV is the most abundant form of the virus and is suitable for mediating transmission between hosts due to its physically robust nature. However, IMV does not spread well in a host due to its susceptibility

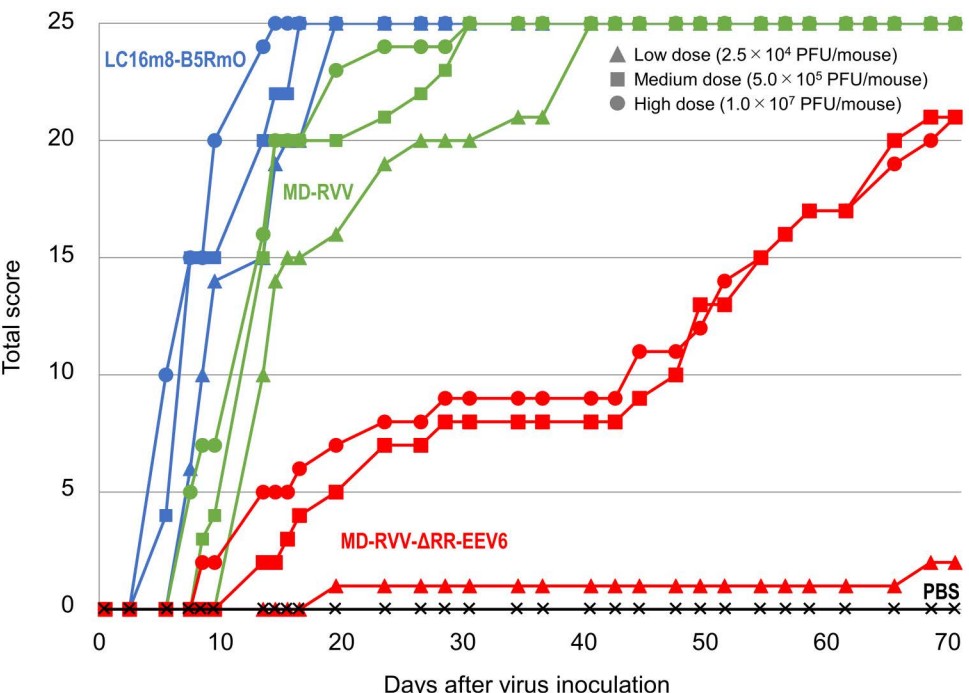

**Fig 8. Virus symptom score changes in immunodeficient mice intravenously administered modified viruses.**
The modified viruses LC16m8-B5RmO (blue), MD-RVV (green), and MD-RVV-ΔRR-EEV6 (red) were intravenously administered once to five immunodeficient mice in each group at doses of $2.5 \times 10^4$ (triangle marks), $5.0 \times 10^5$ (square marks), and $1.0 \times 10^7$ (round marks) PFU/mouse. As a control group (black; cross marks), the same volume of phosphate-buffered saline was administered. Scores represent the sum of virus symptom scores in each treatment group. The definition of an individual's viral symptom score is as follows: 0: no symptoms; 1: 1 or 2 tail lesion(s); 2: 3 or 4 tail lesions; 3: more than 4 tail lesions, a rough coat, piloerection; 4: dyspnea, moribund; 5: death.

to complements and antibodies. IEV acts as an intermediate between IMV and CEV/EEV, ensuring the uptake of EEV-specific proteins, transporting virions to the cell surface using microtubules, and covering IMV particles with additional membranes and host proteins to decrease susceptibility to antibodies and complements. CEV is required to induce the formation of actin tails from the lower part of virions on the cell surface and promote the efficient intercellular transport of the virus. EEV is eventually released from the cell surface and mediates the spread of infection in the host [29]. A virus preparation in the final EEV form is desirable for the oncolytic vaccinia virus drugs used for intravenous administration. However, since the morphology of virions continuously changes during the life cycle of the virus, not only EEV released into the culture supernatant during the virus culture, but also virions, such as IMV, leak from cells destroyed by viral proliferation and become mixed. In terms of productivity and physical stability, it is difficult to produce EEVs in isolation. Therefore, the modification of EEV-associated molecules by viral genome editing may increase the efficacy and productivity of oncolytic virus preparations containing different virion forms. In the present study, the EEV-associated gene sequences present in the viral genomes of MD-RVV and MD-RVV-ΔRR were replaced with the corresponding sites of the IHD-J strain, and their productivity in HeLa cells was examined. A34R, an EEV-associated protein, was previously shown to be important for EEV infectivity [10]. As shown in Fig 4, 48 hours after the inoculation, the productivity of the virus in which 6 genes other than A34R (MD-RVV-EEV6 and MD-RVV-ΔRR-EEV6) were replaced with the sequence of the IHD-J strain was higher

**Fig 9. Typical symptoms in mice treated with modified viruses.** Photographs of viral symptoms in representative mice from each group 15 days after the administration of modified viruses are shown. The modified viruses LC16m8-B5RmO, MD-RVV, and MD-RVV-ΔRR-EEV6 were intravenously administered once to five immunodeficient mice in each group at doses of $5.0 \times 10^5$ PFU/mouse. As a control group, the same volume of phosphate-buffered saline (PBS) was administered.

than the virus in which only A34R was replaced and MD-RVV. The viruses in which all seven EEV-related genes were replaced with sequences from the IHD-J strain (MD-RVV-EEV7 and MD-RVV-ΔRR-EEV7) showed the highest productivity. Increases in production capacity in HeLa cells due to the replacement of these EEV-related genes were not markedly affected by the RNR deletion, suggesting that they are involved in viral infectivity and maturation rather than viral replication capacity. The amount of the virus produced by MD-RVV-ΔRR-EEV7 in the culture supernatant was markedly higher than that of MD-RVV-ΔRR-A34R (Fig 5A). The amount of the virus produced by MD-RVV-ΔRR-EEV6 in cells was significantly higher than that by MD-RVV-ΔRR-A34R and similar to that by MD-RVV-ΔRR-EEV7 (Fig 5B). These results revealed that not only A34R, which is known to be important for EEV production, but

also other proteins related to EEV formation play an important role in increasing infectious virus production.

The intranasal inoculation of BALB/c mice with the A34R deletion mutant of vaccinia virus strain WR has been reported to reduce the viral virulence of the parental strain [10]. We intravenously administered MD-RVV-ΔRR-EEV6, in which only the A34R gene was derived from LC16m8 and the genes for the other six EEV-associated proteins were derived from the IHD-J and IHD-W strain to SCID mice and evaluated its safety. Three doses of LC16m8-B5RmO, MD-RVV, and MD-RVV-ΔRR-EEV6 were intravenously administered to five SCID mice in each group. Under these experimental conditions, the administration of LC16m8-B5RmO and MD-RVV caused all animals to die after 19 and 40 days, respectively, even at low doses. On the other hand, with the administration of MD-RVV-ΔRR-EEV6, 3 out of 5 mice survived 71 days even at the high dose (Fig 6). Similar results were observed for changes in body weight (Fig 7) and symptom scores (Fig 8) after the virus challenge in these mice. *In vivo* safety results for each virus using SCID mice were consistent with *in vitro* cytotoxicity results against NHDF (Fig 3B). The intravenous administration of MD-RVV-ΔRR-EEV7 was not performed in the present study due to limitations in the number of animals simultaneously examined and animal welfare ethics. However, MD-RVV-ΔRR-EEV7 exhibited similar low cytotoxicity to NHDF *in vitro* as RVV-ΔRR-EEV6 and is expected to be safe even when intravenously administered.

Since the majority of passenger mutations in cancer patients are tumor-specific, personalized cancer vaccines may have greater success. Although various mRNA, DNA, or peptide-based cancer vaccines have been investigated, challenges include the analysis of patient-specific neoantigens and major histocompatibility complex (MHC) class I epitopes as well as adjuvant selection [30]. Previous studies examined novel cancer vaccines using fixed or inactivated autologous tumor tissue [31,32]. On the other hand, cancer cell fragments destroyed by oncolytic viruses in patients are expected to induce anti-tumor immune responses specific to autologous cancers. Virotherapy against cancer may function for antitumor immune responses by presenting autologous tumor-associated antigens and releasing inflammatory signals. The combination of oncolytic viruses with immune modulators may enhance the efficacy of both immune and viral therapies. Additionally, the genetic engineering of oncolytic viruses permitted the local expression of immune therapeutics, thereby reducing related toxicities [33].

As mentioned above, this study focused on constructing a novel recombinant oncolytic vaccinia virus and confirming its safety *in vitro* and *in vivo*. In the future, it is expected that these viruses will be used for cancer immunological analyses as well as cancer model animal and clinical research. Vaccinia virus is an excellent viral vector because it has a relatively large viral genome that allows for the insertion of whole genes for multiple proteins, and since it replicates in the cytoplasm of infected cells and minimizes the risk of genetic material being integrated into the host genome. Oncolytic vaccinia virus is a type of attenuated virus whose growth is severely restricted in normal cells. The use of each oncolytic virus as a viral vector is advantageous for inducing antitumor immune responses [34]. The oncolytic vaccinia virus used in the present study may also be used as a gene insertion vector to arm immunostimulatory functions and is expected to be developed as a systemic oncolytic immunotherapeutic drug for patients with metastatic cancer.

## Acknowledgments

The authors are grateful to Dr. Yasushi Kawaguchi from Tokyo University for providing expertise to construct recombinant viruses using BAC technology and sharing pEPkan-S and GS1783. We also thank Y. Kamizuru, S. Etou, K. Gotou, A. Tsune, M. Shinmura, Y. Kanadome,

and S. Kuriyama for constructing the recombinant viruses and obtaining evaluation data as lab technicians.

## Author contributions

**Conceptualization:** Masashi Sakaguchi, Toshio Murakami.

**Data curation:** Go Okita.

**Formal analysis:** Go Okita.

**Investigation:** Go Okita, Kiyotaka Suenaga, Masashi Sakaguchi, Toshio Murakami.

**Methodology:** Go Okita, Kiyotaka Suenaga, Masashi Sakaguchi, Toshio Murakami.

**Writing – original draft:** Go Okita, Toshio Murakami.

**Writing – review & editing:** Go Okita, Kiyotaka Suenaga, Masashi Sakaguchi, Toshio Murakami.

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
