## [Decision Letter · Decision Letter 0]

14 Nov 2023

PONE-D-23-30862Oncolytic vaccinia virus with multiple gene

modifications increases safety while maintaining

proliferative potential in cancer cellsPLOS ONE

Dear Dr. Okita,

Thank you for submitting your manuscript to PLOS ONE. After careful consideration, we feel that it has merit but does not fully meet PLOS ONE’s publication criteria as it currently stands. Therefore, we invite you to submit a revised version of the manuscript that addresses the points raised during the review process.

We look forward to receiving your revised manuscript.

Kind regards,

Milad Zandi, Ph.D.

Academic Editor

PLOS ONE

[We would like to reject Dr. Takafumi Nakamura at Tottori University as a reviewer due to competing financial interests. ]. 

4. We note that Figure 2 and 9 in your submission contain copyrighted images. All PLOS content is published under the Creative Commons Attribution License (CC BY 4.0), which means that the manuscript, images, and Supporting Information files will be freely available online, and any third party is permitted to access, download, copy, distribute, and use these materials in any way, even commercially, with proper attribution. For more information, see our copyright guidelines: http://journals.plos.org/plosone/s/licenses-and-copyright.

a. You may seek permission from the original copyright holder of Figure 2 and 9 to publish the content specifically under the CC BY 4.0 license. 

Reviewers' comments:

Reviewer's Responses to Questions

**Comments to the Author**

1. Is the manuscript technically sound, and do the data support the conclusions?

Reviewer #1: Yes

Reviewer #2: No

2. Has the statistical analysis been performed appropriately and rigorously? 

Reviewer #1: No

Reviewer #2: Yes

3. Have the authors made all data underlying the findings in their manuscript fully available?

Reviewer #1: Yes

Reviewer #2: No

4. Is the manuscript presented in an intelligible fashion and written in standard English?

Reviewer #1: Yes

Reviewer #2: Yes

5. Review Comments to the Author

Reviewer #1: Dear author,

The strength of the paper is well structured, at all the work is of certain significance. The paper provides a very powerful message about the oncolytic vaccinia virus which used in this study as a gene insertion vector to arm immunostimulatory functions and is expected to be developed as a systemic oncolytic immunotherapeutic drug for patients with metastatic cancer. However, the study was needed to some modifications as following to be ready for publications. Please address the following comments in your consideration:

1-All figures in the manuscript will need to modifications; it is hesitated and not clear.

2-The manuscript will need to the statistical evaluation.

Reviewer #2: Despite the authors' efforts, the study design is poor, and the study lacks studies such as cancer cell apoptosis and western blotting, which are necessary to confirm the results. For such studies, the proteins of the apoptotic pathway and their signaling should be examined. In addition, the topic of this manuscript is not a priority and the manuscript is boring, for example, there is no need to write such a long abstract.

6. PLOS authors have the option to publish the peer review history of their article (what does this mean? ). If published, this will include your full peer review and any attached files.

**Do you want your identity to be public for this peer review?** For information about this choice, including consent withdrawal, please see our Privacy Policy .

Reviewer #1: No

Reviewer #2: No

---

## [Author Response · Author response to Decision Letter 1]

27 Dec 2023

Dear Ms. Syrra Mica Hilvano,

Thank you for confirming with reviewer #1 the meaning of his (her) comment the other day.

We didn't get a response from reviewer #1, but we understood that the problem was in the way the figures were created.

We re-created all figures and verified that the "Image Problems" were resolved using the Preflight Analysis and Conversion Engine (PACE) digital diagnostic tool.

We would like to inform you that we have addressed this issue and submitted a revised version of the figures.

Best regards,

Go Okita

---

## [Decision Letter · Decision Letter 1]

15 Jan 2024

PONE-D-23-30862R1Oncolytic vaccinia virus with multiple gene

modifications increases safety while maintaining

proliferative potential in cancer cellsPLOS ONE

Dear Dr. Okita,

Thank you for submitting your manuscript to PLOS ONE. After careful consideration, we feel that it has merit but does not fully meet PLOS ONE’s publication criteria as it currently stands. Therefore, we invite you to submit a revised version of the manuscript that addresses the points raised during the review process.

We look forward to receiving your revised manuscript.

Kind regards,

Milad Zandi, Ph.D.

Academic Editor

PLOS ONE

Journal Requirements:

Additional Editor Comments:

- The design of this study is good, but the big problem is why the authors considered SCID mice model and didn’t experiment on murine cancer model. It is more valuable if the authors investigate immune responses in immunocompetent mice model to evaluate modified vaccinia inoculation into in-vivo models for evaluating the potential of these modified vaccinia to induce antitumor immune responses.

- The abstract doesn’t include a conclusion relevant to the finding. Please reorganize the abstract according to the submission guideline.

- The title needs to mention more specifically, for example, indicate the unique modifications for the oncolytic vaccinia virus.

- In Fig. 9, the resolution of mice photographs is poor and viral symptoms induced by each modified virus are not indicated well.

- Please indicate labels for each bar chart for normal and Hela cells in Fig. 3.

- In Fig. 2, mock Hela cell is not shown to compare with other modified viruses infected Hela.

- There is a lack of statistical analysis section in the manuscript.

- In the method section, the authors didn’t mention what tests (MTT or ...) were used for evaluating oncolytic efficacy on various cells. Please explain about the cell viability assay you used in the method section and determine the cut-off for PFU/cell for determining notable cytotoxicity in normal and cancer cell line.

Reviewers' comments:

Reviewer's Responses to Questions

**Comments to the Author**

1. If the authors have adequately addressed your comments raised in a previous round of review and you feel that this manuscript is now acceptable for publication, you may indicate that here to bypass the “Comments to the Author” section, enter your conflict of interest statement in the “Confidential to Editor” section, and submit your "Accept" recommendation.

Reviewer #1: All comments have been addressed

Reviewer #3: All comments have been addressed

2. Is the manuscript technically sound, and do the data support the conclusions?

Reviewer #1: Yes

Reviewer #3: Partly

3. Has the statistical analysis been performed appropriately and rigorously? 

Reviewer #1: Yes

Reviewer #3: No

4. Have the authors made all data underlying the findings in their manuscript fully available?

Reviewer #1: Yes

Reviewer #3: Yes

5. Is the manuscript presented in an intelligible fashion and written in standard English?

Reviewer #1: Yes

Reviewer #3: Yes

6. Review Comments to the Author

Reviewer #1: The modifications made by the authors made the research paper ready for international publication, and this is from my professional point of view.

Reviewer #3: - The design of this study is good, but the big problem is why the authors considered SCID mice model and didn’t experiment on murine cancer model. It is more valuable if the authors investigate immune responses in immunocompetent mice model to evaluate modified vaccinia inoculation into in-vivo models for evaluating the potential of these modified vaccinia to induce antitumor immune responses.

- The abstract doesn’t include a conclusion relevant to the finding. Please reorganize the abstract according to the submission guideline.

- The title needs to mention more specifically, for example, indicate the unique modifications for the oncolytic vaccinia virus.

- In Fig. 9, the resolution of mice photographs is poor and viral symptoms induced by each modified virus are not indicated well.

- Please indicate labels for each bar chart for normal and Hela cells in Fig. 3.

- In Fig. 2, mock Hela cell is not shown to compare with other modified viruses infected Hela.

- There is a lack of statistical analysis section in the manuscript.

- In the method section, the authors didn’t mention what tests (MTT or ...) were used for evaluating oncolytic efficacy on various cells. Please explain about the cell viability assay you used in the method section and determine the cut-off for PFU/cell for determining notable cytotoxicity in normal and cancer cell line.

7. PLOS authors have the option to publish the peer review history of their article (what does this mean? ). If published, this will include your full peer review and any attached files.

**Do you want your identity to be public for this peer review?** For information about this choice, including consent withdrawal, please see our Privacy Policy .

Reviewer #1: No

Reviewer #3: **Yes: ** Zahra Heydarifard

---

## [Author Response · Author response to Decision Letter 2]

26 Feb 2024

Dear Dr. Zandi:

Thank you for your email dated January 15, 2024, enclosing the reviewers’ comments. We appreciate your kind suggestion about resubmission and the constructive comments made by you and the reviewers. We have carefully reviewed the comments and have revised the manuscript accordingly. Our responses are given in a point-by-point manner below. Other minor corrections are reflected in the ‘Revised Manuscript with Track Changes’ file.

We hope the revised version is now suitable for publication and look forward to hearing from you in due course.

Go Okita

KM Biologics Co., Ltd.

---

## [Decision Letter · Decision Letter 2]

23 Jun 2024

PONE-D-23-30862R2A novel oncolytic vaccinia virus with multiple gene modifications involved in viral replication and maturation increases safety for intravenous administration while maintaining proliferative potential in cancer cellsPLOS ONE

Dear Dr. Okita,

Thank you for submitting your manuscript to PLOS ONE. After careful consideration, we feel that it has merit but does not fully meet PLOS ONE’s publication criteria as it currently stands. Therefore, we invite you to submit a revised version of the manuscript that addresses the points raised during the review process. In particular:

Both reviewers have pointed out the lack of statistics in critical pieces of data. This was previously requested and not addressed. Statistical analysis on the designated data will need to be done before the manuscript can be accepted.Both reviewers found the figures to be sub-standard. Please fix accordingly.One of the reviewers has pointed out that the actual viruses created using the Bac system were not genetically verified. This is a critical step to be sure the viruses used in the manuscript have the genotypes desired.

We look forward to receiving your revised manuscript.

Kind regards,

Brian M. Ward, Ph.D.

Academic Editor

PLOS ONE

Reviewers' comments:

Reviewer's Responses to Questions

**Comments to the Author**

1. If the authors have adequately addressed your comments raised in a previous round of review and you feel that this manuscript is now acceptable for publication, you may indicate that here to bypass the “Comments to the Author” section, enter your conflict of interest statement in the “Confidential to Editor” section, and submit your "Accept" recommendation.

Reviewer #3: All comments have been addressed

Reviewer #4: (No Response)

Reviewer #5: (No Response)

2. Is the manuscript technically sound, and do the data support the conclusions?

Reviewer #3: Yes

Reviewer #4: Partly

Reviewer #5: Yes

3. Has the statistical analysis been performed appropriately and rigorously? 

Reviewer #3: Yes

Reviewer #4: No

Reviewer #5: No

4. Have the authors made all data underlying the findings in their manuscript fully available?

Reviewer #3: Yes

Reviewer #4: Yes

Reviewer #5: Yes

5. Is the manuscript presented in an intelligible fashion and written in standard English?

Reviewer #3: Yes

Reviewer #4: Yes

Reviewer #5: Yes

6. Review Comments to the Author

Reviewer #3: (No Response)

Reviewer #4: I have not previously reviewed an earlier version of this manuscript.

The paper describes a solid piece of work and highlights the growth advantages gained by swapping the EEV genes from IHD-J into LC18m8, which was originally attenuated by a B5R frameshift mutation. It also confirms many of the growth properties of deleting F4L. The differences between viruses are not that great (a few fold in most cases), but better yields are always helpful and systemic spread might offer some advantages tackling metastatic cancers.

From an experimental perspective my only major concern is that the authors have not genome sequenced their viruses. This is a concern because if one uses a homologous poxvirus to reactivate another poxvirus, there is a significant risk of recombination between the transfected and infecting DNA. Simply screening for loss of the helper-virus encoded GFP plus some PCR won't necessarily confirm the identity of a virus as there may be short patches of one virus introgressed into another. Maybe the PCR data is good enough, but given how easy it is to sequence poxviruses nowadays, I'd strongly suggest that the authors use whole genome sequencing to confirm the identity of each strain. The method will also detect randomly scattered mutations arising during the various manipulations and which are not detected by PCR.

BTW, it would be good to cite (PMID: 11570497) as it provides an earlier example of using a vaccinia to reactivate another vaccinia albeit with a different trick to the method.

As a general comment I found the figures poorly drawn and organized. For example, to decipher what's shown in Figure 4 the reader has to first figure out what virus the letters a-e represent (using the figure legend), and then since that nomenclature is not so easily deciphered, turn to Figure 1. A rethink of how best to plot the data with a reader's convenience in mind is needed. For example it would be much clearer if each of the 4 time points in Fig 4 were regrouped by virus with the virus's name listed underneath the 4 collected time points. I'd also note that Figure 6 is a Kaplan-Meier plot, but it's not correctly drawn that way nor properly analyzed using statistics.

Only Fig 3 has any attached statistics, in fact all of graphs need further analysis. This was requested previously by another reviewer, but has not been properly addressed in the revision. (The authors might want to purchase a copy of Prism which can both analyze the data and graph it along with the calculated statistics.)

Some of the speculation in the discussion concerns matters previously investigated in ref. 13, but not here properly cited. This includes the interactions with cellular RR subunits. Whether a delta F4L virus can be used as an oncolytic virus has also been tested (see DOI: 10.1158/2326-6066.CIR-19-0703) but is again not cited.

Other

p12 (line 87) - "highly homologous". Homology is a word like "dead", you are dead or not dead, but can't be "highly dead". Use "similar" where a degree of similarity needs to be mentioned.

p12 (line 100) - vaccinia encodes two genes for the RR.

p17 (line 172) - Suggest indicating that black arrows are genes from IHD-J

pp17-19 - Suggest condensing and/or deleting much of the description of plasmid and BACmid construction. These are standard methods and rarely included in M&M's nowadays.

p. 22 (line 269) - coli not Coli. Not sure of PLoS One policy on italics, but the restriction enzymes aren't properly italicized.

p. 32 (line 426) - Some journals don't permit "data not shown"

p. 33 (line 442-443) - It's much easier to do the math for the reader. That is 8x10^4 and 4x10^5 PFU. The MOI must also be reported based on cell counts at the time. The preferred approach would be to plot cytotoxicity over a range of MOI's, not just pick a narrow (5-fold) window where the effects are perhaps most exaggerated.

p. 33 (line 447) - There's something missing where it says "(data labels)"

p. 42 (line 597-598) - See comment regarding lines 442-443.

Figures 4 and 5. The Y axis should be labelled as x10^-4 (or -6 or -7). The label indicates what math operation was applied to the actual data to create the axis numbering. In such experiments the titers (e.g. 4x10^4 PFU/mL) would have been multiplied by 10^-4 to plot them.

I'm not convinced Figure 9 adds anything to the paper as most expert readers would be familiar with mouse pathology.

Reviewer #5: There is a lack of statistical analysis, particularly as it relates to Figure 4 and 5. It looks like there may be statistical differences. Were statistics conducted? If so, statistical analysis should be included and described. If not, why were statistical analysis not completed?

Figure 9: the resolution of the mouse photographs is poor. The viral symptoms that the authors note are present, need to be indicated more clearly. It is not clear what exactly we are looking at.

In general, figures could be better labeled so as it make it easier for the reader to follow what is being shown. In particular, a legend for the different colors and shapes should be provided within the figure for Figures 6, 7, and 8. While it is laid out in the figure legend, it is difficult to follow. Furthermore, labeled data bars with the virus being used rather than with letters, would make it easier to follow.

line 18 in the abstract: consider changing "high safety" to "improved safety"

Reference to normal cells is present throughout the manuscript. What is the definition of a "normal" cell? This is not well-defined.

There are a lack of citations throughout the Introduction where citations should be present. For example, line 63, 66. Ensure citations are present where prior work is being cited or referenced.

Line 345 and 350: "centrifugal supernatant" is not clear. Modify the wording. Is this supposed to read, supernatant centrifuged?

Line 415: how do you define "many smallpox lesions"? Is there a number?

Line 463: Reference to "data labels" is unclear. Please be more specific.

Line 545: This statement "as an animal symptom due to virus administration" is unclear.

7. PLOS authors have the option to publish the peer review history of their article (what does this mean? ). If published, this will include your full peer review and any attached files.

**Do you want your identity to be public for this peer review?** For information about this choice, including consent withdrawal, please see our Privacy Policy .

Reviewer #3: **Yes: ** zahra heydarifard

Reviewer #4: No

Reviewer #5: No

---

## [Author Response · Author response to Decision Letter 3]

6 Aug 2024

Dear Dr. Ward:

Thank you for your email dated June 24, 2024, enclosing the reviewers’ comments. We appreciate your kind suggestion about resubmission and the constructive comments made by you and the reviewers. We have carefully reviewed the comments and have revised the manuscript accordingly. Our responses are given in a point-by-point manner below. Other minor corrections are reflected in the ‘Revised Manuscript with Track Changes’ file.

We hope the revised version is now suitable for publication and look forward to hearing from you in due course.

Go Okita

KM Biologics Co., Ltd.

---

## [Decision Letter · Decision Letter 3]

8 Sep 2024

A novel oncolytic vaccinia virus with multiple gene modifications involved in viral replication and maturation increases safety for intravenous administration while maintaining proliferative potential in cancer cells

PONE-D-23-30862R3

Dear Dr. Okita,

We’re pleased to inform you that your manuscript has been judged scientifically suitable for publication and will be formally accepted for publication once it meets all outstanding technical requirements.

Kind regards,

Brian M. Ward, Ph.D.

Academic Editor

PLOS ONE

Additional Editor Comments (optional):

The following items need to be fixed before the paper can be published.

The following items need to be fixed:

Please provide citations for the statements on:

Line 111

Line 187. Citation 19 is not a journal article. Please be sure there is a citation that accurately describes the “BAC system”.

Line 403

Line 573-575

Line 575-580

Line 586-589

Line 589-560

This appears to be the creation and first use of the described BACmid. Therefore, a better description of its creation is required.

Much of the BACmid construction methods is “uncited” and poorly described. For instance, what is “pUC-VVTK-BAC-EGFP” and pCAGGS-Cre? Where did they come from and what do they do? Why was Cre used? What is LC16m8-BACmid and where did it come from? A diagram of the circular BACmid showing the relevant parts would be very helpful in understanding the construct and how it was made. It should be stated if the starting BACmid that was constructed was sequenced. If not, it should be stated that the genotype of it and the resulting viruses are unverified.

Similarly, the EEV protein cassettes section (lines 225-236) is not well written and confusing. Why was kanamycin resistance added and where? Why were they “treated with restriction enzymes”? Which enzymes and why?

Please describe the selective culturing (line 241).

PRKs are not adequately described.

The sentence on line 189 is a run-on sentence.

Line 246-247 is confusing. What is BACgfp and how was it removed?

It is unclear what is meant by “we increased the productivity of the oncolytic virus.” Line 125, and Line 604.

The statement “…it was shown that any virus strain can be used as a helper virus” is not accurate (Line 134). Please modify to be accurate.

Were the cells on line 317 stained before counting plaques?

The statement on line 612 should state that the results confirm previous results that EEV proteins play important roles……

Vaccinia virus does not cause smallpox lesions. Please rename.

Please use the actual strain names and not the terms “parent strains” and “modified virus” (Line 345)

Terms such as “basic virus” (Line 358), “genome gene” (Line 365) and “completed viruses” (Line 368) are vague. Consider revising.

All infections should have explicit MOIs stated.

The conclusions stated in the paragraph starting on line 535 should include the caveat that the genomes of the viruses created were not sequenced and therefore you cannot rule out aberrant genotypes contributing to the observed phenotypes.

It should be explicitly stated how many times each experiment was conducted.

Statistical significance should be carried out for Fig. 4 and 5.

Reviewers' comments:

Reviewer's Responses to Questions

**Comments to the Author**

1. If the authors have adequately addressed your comments raised in a previous round of review and you feel that this manuscript is now acceptable for publication, you may indicate that here to bypass the “Comments to the Author” section, enter your conflict of interest statement in the “Confidential to Editor” section, and submit your "Accept" recommendation.

Reviewer #4: All comments have been addressed

2. Is the manuscript technically sound, and do the data support the conclusions?

Reviewer #4: (No Response)

3. Has the statistical analysis been performed appropriately and rigorously? 

Reviewer #4: (No Response)

4. Have the authors made all data underlying the findings in their manuscript fully available?

Reviewer #4: (No Response)

5. Is the manuscript presented in an intelligible fashion and written in standard English?

Reviewer #4: (No Response)

6. Review Comments to the Author

Reviewer #4: (No Response)

7. PLOS authors have the option to publish the peer review history of their article (what does this mean? ). If published, this will include your full peer review and any attached files.

**Do you want your identity to be public for this peer review?** For information about this choice, including consent withdrawal, please see our Privacy Policy .

Reviewer #4: No

---

## [Editor Report · Acceptance letter]

PONE-D-23-30862R3

PLOS ONE

Dear Dr. Okita,

I'm pleased to inform you that your manuscript has been deemed suitable for publication in PLOS ONE. Congratulations! Your manuscript is now being handed over to our production team.

Kind regards,

on behalf of

Dr. Brian M. Ward

Academic Editor

PLOS ONE